# Predictive Biomarkers of Response to Neoadjuvant Chemotherapy in Breast Cancer: Current and Future Perspectives for Precision Medicine

**DOI:** 10.3390/cancers14163876

**Published:** 2022-08-11

**Authors:** Françoise Derouane, Cédric van Marcke, Martine Berlière, Amandine Gerday, Latifa Fellah, Isabelle Leconte, Mieke R. Van Bockstal, Christine Galant, Cyril Corbet, Francois P. Duhoux

**Affiliations:** 1Department of Medical Oncology, King Albert II Cancer Institute, Cliniques Universitaires Saint-Luc, Avenue Hippocrate 10, 1200 Brussels, Belgium; 2Breast Clinic, King Albert II Cancer Institute, Cliniques Universitaires Saint-Luc, Avenue Hippocrate 10, 1200 Brussels, Belgium; 3Institut de Recherche Expérimentale et Clinique (IREC), Pole of Medical Imaging, Radiotherapy and Oncology (MIRO), Université Catholique de Louvain (UCLouvain), 1200 Brussels, Belgium; 4Department of Gynecology, King Albert II Cancer Institute, Cliniques Universitaires Saint-Luc, Avenue Hippocrate 10, 1200 Brussels, Belgium; 5Institut de Recherche Expérimentale et Clinique (IREC), Pole of Gynecology (GYNE), Université Catholique de Louvain (UCLouvain), 1200 Brussels, Belgium; 6Department of Radiology, Cliniques Universitaires Saint-Luc, Avenue Hippocrate 10, 1200 Brussels, Belgium; 7Department of Pathology, Cliniques Universitaires Saint-Luc, Avenue Hippocrate 10, 1200 Brussels, Belgium; 8Institut de Recherche Expérimentale et Clinique (IREC), Pole of Pharmacology and Therapeutics (FATH), Université Catholique de Louvain (UCLouvain), 1200 Brussels, Belgium

**Keywords:** breast cancer, neoadjuvant chemotherapy, biomarkers, predictive factors, intratumoral heterogeneity

## Abstract

**Simple Summary:**

Despite the increased use of neoadjuvant chemotherapy in the early setting of breast cancer, there is a clinical need for predictive markers of response in daily practice. In the era of precision medicine and personalized treatment, new predictive markers that enable the better selection of patients for specific therapies are required. In this review, we describe the current knowledge about the molecular biomarkers used for clinical decision making for patients with breast cancer. We also report how microenvironment-driven intratumoral heterogeneity may influence the validation of biomarkers useful for precision medicine. We provide an overview of promising biomarkers, including pathological markers, genetic signatures, radiological techniques and liquid biopsies, with a great potential to be implemented in routine clinical practice. Finally, we discuss the use of relevant pre-clinical models of breast cancer to integrate microenvironmental specificities in order to identify and validate reliable biomarkers of (non-)response to neoadjuvant chemotherapy.

**Abstract:**

Pathological complete response (pCR) after neoadjuvant chemotherapy in patients with early breast cancer is correlated with better survival. Meanwhile, an expanding arsenal of post-neoadjuvant treatment strategies have proven beneficial in the absence of pCR, leading to an increased use of neoadjuvant systemic therapy in patients with early breast cancer and the search for predictive biomarkers of response. The better prediction of response to neoadjuvant chemotherapy could enable the escalation or de-escalation of neoadjuvant treatment strategies, with the ultimate goal of improving the clinical management of early breast cancer. Clinico-pathological prognostic factors are currently used to estimate the potential benefit of neoadjuvant systemic treatment but are not accurate enough to allow for personalized response prediction. Other factors have recently been proposed but are not yet implementable in daily clinical practice or remain of limited utility due to the intertumoral heterogeneity of breast cancer. In this review, we describe the current knowledge about predictive factors for response to neoadjuvant chemotherapy in breast cancer patients and highlight the future perspectives that could lead to the better prediction of response, focusing on the current biomarkers used for clinical decision making and the different gene signatures that have recently been proposed for patient stratification and the prediction of response to therapies. We also discuss the intratumoral phenotypic heterogeneity in breast cancers as well as the emerging techniques and relevant pre-clinical models that could integrate this biological factor currently limiting the reliable prediction of response to neoadjuvant systemic therapy.

## 1. Introduction

Breast cancer remains one of the most prevalent cancers in women, with 2,261,419 new cases and 684,996 deaths worldwide in 2020, despite major improvements in terms of prevention, diagnosis and treatment [1]. In current clinical practice, breast cancers are classified in five subtypes based on the expression of hormone receptors (estrogen and/or progesterone receptors (ER and/or PgR)), the overexpression of epidermal growth factor receptor 2 (HER2/Neu) and the percentage of tumor cells expressing Ki-67. These subtypes comprise luminal A, luminal B, HR+ HER2-positive, HR- HER2-positive and triple-negative breast cancers (TNBC) [2]. The majority of patients are diagnosed with early-stage disease, while 3–10% of patients are diagnosed with de novo metastatic breast cancer [3]. Although most early breast cancers are curable with the current treatment options, up to 20% of patients will relapse within 10 years. At present, treatment decisions in both the early and the metastatic settings depend on the immunohistopathological classification, with adaptation of the chemotherapy regimen based on the surrogate molecular subtype (e.g., the addition of anti-HER2 monoclonal antibodies in the HER2+ subtype, the addition of carboplatin in the early triple negative subtype, adjuvant hormonotherapy in the HR+ subtypes, etc.) [2,4]. (Table 1) The implementation of neoadjuvant chemotherapy (NAC) as the current standard of care for patients with high-risk early-stage or locally advanced breast cancer is one of the major changes in the evolving breast cancer landscape [2]. The high-risk breast cancers concerned by this change in the treatment paradigm are mainly TNBC and HER2-positive tumors but also include hormone-receptor-positive (HR+) cancers larger than 2 cm and/or with axillary lymph node involvement. While providing the same overall survival (OS) and disease-free survival (DFS) as adjuvant chemotherapy, NAC has several advantages, such as: allowing for more conservative surgeries by reducing the tumor size and down-staging the lymph node status, assessing the sensitivity of the tumor to chemotherapeutic agents and eradicating micro-metastases but also adding the possibility of escalating treatment with adjuvant drugs in case of residual disease, a feature of worse prognosis [5,6,7]. Despite this, NAC also has disadvantages, including: drug-related side effects (see Table 1), the postponement of surgery in some cases (e.g., the postponement of NAC due to side effects resulting in a longer delay before surgery), difficulties in healing after surgery and disease progression that may occur during treatment [8] (Figure 1).

Pathological complete response (pCR) after NAC has been validated as a prognostic factor with an improved event-free survival (EFS) and OS when the tumor is eradicated from both breast and lymph nodes (corresponding to the pathologic stage ypT0 ypN0 or ypTis ypN0) [9]. This statement is particularly true in triple-negative tumors and HER2-positive, hormone-receptor-negative (HER2+ HR-) breast cancers [10]. Achieving pCR is therefore one of the main objectives of NAC but occurs only in a minority of patients, with 30–50% of pCR in TNBC, 50–80% in HER2-positive breast cancer and 5–20% in high-grade luminal cancers. [11,12]. Predicting the response to NAC in early-stage breast cancer represents a challenge for several reasons: (i) the surrogate molecular subtype is not sufficient in and of itself to correctly classify patients since outcomes can be very heterogeneous within each subtype, and (ii) the intra-tumoral heterogeneity is incompletely assessed in routine (e.g., diagnosis biopsy, radiological examination) [13,14]. Importantly, the prediction of response to NAC is of interest for better patient selection and tailoring (by escalating or de-escalating) chemotherapy regimens. The better selection of patients for NAC could reduce the likelihood of facing residual disease, which is known to be more resistant than the primary tumor [15]. It is of interest for clinical trials to test and validate new drugs in the neoadjuvant setting, as is currently being done in the I-SPY 2 study [16].

In this review, we describe the current knowledge about the molecular biomarkers used for clinical decision making for patients with breast cancer and highlight the future perspectives that could lead to the better prediction of response to NAC in early breast cancer patients, focusing on the current biomarkers used in clinical decision making and the different gene signatures that have recently been proposed for patient stratification and the prediction of response to therapies. We also detail the intratumoral phenotypic heterogeneity in breast cancers and discuss the emerging techniques and relevant pre-clinical models that could integrate this biological factor currently limiting the reliable prediction of response to neoadjuvant systemic therapy in this disease.

## 2. Breast Cancer Subtypes and Intratumoral Heterogeneity

### 2.1. Molecular Classification and Intrinsic Subtypes

Over the past 20 years, several molecular classifications have been determined by genomic and transcriptomic clustering in order to better understand the intratumoral heterogeneity in breast cancer. However, these intrinsic subtypes have not yet supplanted the surrogate molecular subtype (determined by immunohistochemistry) in clinical practice for therapeutic decision making. In 2000, Perou et al. analyzed 65 surgical pieces of breast cancers from 42 individuals and identified 4 intrinsic subtypes by gene expression analysis: luminal-like, basal-like, normal-like and HER2-enriched [17]. Later, the PAM50 classification distinguished the luminal A and luminal B categories within the luminal-like group. Studies have shown that these subgroups differ in both their clinical characteristics and their response to treatment. Prognosis also differs between the intrinsic subtypes, regardless of the immunohistopathological subtype [18,19,20,21,22]. Within the immunohistochemical triple negative subtype, Prat et Perou later highlighted two intrinsic subtypes: the claudin-low and the basal-like subtypes, the claudin-low being associated with poorer prognosis [23,24]. The heterogeneity of the triple negative disease is nevertheless more complex, and in 2011, Lehmann et al. described seven subtypes (TNBCtype): basal-like 1 (BL1), basal-like (BL2), immunomodulatory (IM), mesenchymal (M), mesenchymal stem-like (MSL), luminal androgen receptor (LAR) and unstable (UNS) [25,26]. In a retrospective study, the response to NAC containing anthracyclines and cyclophosphamide was different among subgroups, with a better pCR rate in the BL1 subgroup (52%) in comparison to the BL2 and LAR subgroups (0% and 10%) [25]. After refining this TNBCtype classification by considering the transcript of normal stroma and immune cells, four subtypes have been largely studied (TNBCtype-4): BL1, BL2, M and LAR [25] (Figure 2).

### 2.2. Intratumoral Heterogeneity in Breast Cancers

Intratumoral heterogeneity (ITH) in breast cancer refers to the diversity found within tumors, existing at several levels [27,28]. ITH includes multiple concepts and principles such as clonal heterogeneity and cell state heterogeneity [14,27,29]. Clonal heterogeneity concerns the phenotypic variability between cells within a tumor depending on spatial and temporal factors, leading to different clones of cells with different sensitivities to treatment. Many spatial factors can influence the clones of cells: hypoxia variance from the center of the tumor to the periphery, angiogenesis, pH variation inside the tumor and interactions with cells from the tumor microenvironment (TME). Depending on their exposure to hypoxia, or other factors, cells will acquire different metabolisms and different sensitivities to treatment. Temporal factors are more related to the sequential exposure of the tumor to different lines of treatment that could lead to a selection of resistant clones within the tumors [14,27]. Cell state heterogeneity is the fact that we can observe cells at different states in a single tumor, with some cells exhibiting stem cell properties (CSC) and others with differentiated properties or progenitor’s properties, altogether forming tumors with different levels of mechanisms of resistance to treatment [27,29]. The heterogeneity found inside each breast cancer has been described as a strong mechanism of resistance to chemotherapy, with the evolution of cell-to-cell interactions but also genetic modifications under treatment pressure [30]. In breast cancer, several molecular subtypes can also be found within a single tumor. Indeed, molecular subtypes are not a static state, and interconversion between subtypes can occur and lead to tumor progression, metastasis or resistance to chemotherapy [29,31]. Several hypotheses could explain molecular subtypes-associated heterogeneity: the selection of clones during therapy, the change in the molecular expression of ER/PR/HER2, the varying microenvironment influence, etc. This heterogeneity has largely been studied in recent years by imaging techniques, pathology and, more recently, by RNA sequencing, with the aim of better predicting the response to NAC [32,33]. Single-cell RNA sequencing (scRNA-seq) was crucial to better understanding the genetic intratumoral heterogeneity. In comparison to bulk RNA sequencing, scRNA-seq has the advantages of analyzing each cell type within a tumor sample and differentiating malignant cells from microenvironmental cells [34,35]. While bulk RNA sequencing reveals global features, scRNA-seq can evaluate the cellular origin and spatial organization within the tumors [36]. In breast cancer, this sequencing technique has demonstrated that the TNBC subtype is much more heterogeneous than other subtypes regarding molecular subtypes but also transcriptional hallmarks activity [31,37]. ScRNA-seq also allowed for the evaluation of resistance mechanisms due to the heterogeneity in breast cancer—notably, in TNBC [38,39].

## 3. Resistance to Neoadjuvant Chemotherapy

The inherent phenomena linked to resistance to neoadjuvant chemotherapy are still under investigation but can be partially explained by several hypotheses related either to the metabolism of the drug or to the tumor cells themselves [40,41].

### 3.1. Drug-Associated Resistance

It has been demonstrated that drug-metabolizing enzymes (DME) could be differentially expressed between cancer cells and normal cells, leading to resistance to chemotherapy. It is well known that the metabolism of most chemotherapy drugs involves cytochrome P450 enzymes (CYP) [42]. Polymorphisms in the *CYP1B1* gene appear to correlate with resistance to taxanes, while CYP2B6 is involved in the metabolism of cyclophosphamide and doxorubicin [43,44]. In one study, CYP2C9*2 heterozygote breast cancer patients had a decrease in the efficacy in neoadjuvant chemotherapy compared to patients with wild-type alleles [45]. Many other CYP enzymes were reported to be associated with the efficacy of neoadjuvant chemotherapy in breast cancer, and this could explain the clinical resistance to certain drugs [42].

Chemotherapy drug concentration is also regulated by the efflux of drugs out of the cells via transmembrane proteins. These proteins include the ATP-binding cassette (ABC) transporter family, in which the P-glycoprotein (P-gp) has already been associated with drug resistance in breast cancer [40]. In TNBC, several related genes are more expressed, such as *ABCC1*, *ABCC11* or *ABCG2*, and those could be involved in chemoresistance to commonly used drugs [41].

### 3.2. Cancer Cell-Associated Resistance

Several studies were dedicated to cancer cells resistance itself, highlighting the potential role of the selection of clones during the treatment and the role of cancer stem cells (CSCs), the change in the expression of genes involved in the DNA damage repair (DDR) system, epithelial-mesenchymal transition (EMT) and the inhibition of apoptosis [15,40].

CSCs are a population of cells with self-renewal properties present in breast cancer tumors, and they have been found in residual tumors after NAC, indicating that these cells are resistant to conventional treatment [46,47]. Moreover, CSCs are found more often in TNBC than in other subtypes and could be involved in the poor survival of this subtype [48,49]. Changes in the genes involved in the DDR system have also been pointed out as a cause of resistance to chemotherapy. Among the incriminated genes, *HORMAD1* could play a role in chemoresistance in TNBC [50]. EMT plays an important role in breast cancer, which could lead to chemotherapy resistance and metastasis [51]. Cells that undergo EMT have common characteristics with CSCs, explaining part of their resistance to chemotherapy. The evasion of apoptosis is another mechanism leading to the resistance to several drugs such as doxorubicin, cyclophosphamide and paclitaxel. The overexpression of factors such as Bcl2, MCL1 or NF-KB has been shown to decrease the sensitivity to chemotherapy [52,53,54].

## 4. Current Biomarkers Used for the Clinical Decision Making of Breast Cancer Patients

### 4.1. Ki-67 before NAC

Ki-67 is a marker of cell proliferation used in clinical practice to assess the aggressiveness of the tumor at the time of diagnosis [55]. Ki-67 is expressed in all the cell cycle phases, with the exception of the G0 phase, and high Ki-67 expression is related to high tumor proliferation and thus a large number of dividing cells [56]. Ki-67 has been evaluated in several studies for its predictive potential, but its use in that indication is still controversial [57]. Nevertheless, in a meta-analysis, Chen et al. [58] analyzed 44 studies and concluded that high Ki-67 expression at diagnosis was associated with increased pCR rates in breast cancer patients treated with anthracycline- and/or taxane-containing NACs. This finding concerned all subtypes of breast cancer and remained significant using different thresholds of Ki-67 (e.g., >15%, >20%, >50%). Even though Ki-67 has not been validated as a predictive marker of pCR, its prognostic value has been largely studied at the moment of diagnosis but also in residual tumors after NAC [55]. In their meta-analysis, Li et al. showed that a high percentage of Ki-67 at diagnosis but also after NAC was correlated with worse outcomes in terms of OS and DFS. Moreover, the absence of a decrease or a low decrease in Ki-67 after NAC were also associated with poorer OS and DFS [55]. One limitation of the available studies that could easily explain the controversial use of Ki-67 as a predictive or prognostic marker is that there is no consensus on the classification of Ki-67 as “high” or “low”. Depending on the studies, cut-offs range from 15% to 50%, sometimes depending on the molecular subtypes [55,58]. Inter-laboratory and inter-observer variability have been pointed out as being responsible for the high variability of this cut-off [59]. In clinical practice, and in accordance with international recommendations, Ki-67 should be interpreted according to local laboratory values, and the cutoff should be determined on a “laboratory-by-laboratory” basis, although the suggested cutoff is 20% [2].

### 4.2. Tumor Size

Tumor size plays a key role in the response to chemotherapy. Livingston-Rosanoff et al. included 38,864 patients between 2010 to 2013 in a retrospective study. These patients underwent NAC and surgery for unifocal lesions ranging in size from cT1 to cT3. This study demonstrated that tumors with a size > 5 cm have a lower chance to achieve pCR, regardless of their immunohistological subtype [60]. This could be explained by the fact that larger tumors have a higher probability of displaying increased heterogeneity, with different populations of cells susceptible to having a variable sensitivity to treatment. Tumor size is therefore a relevant predictive factor of non-response to NAC, but it is not sufficient to predict whether patients will achieve a pCR or not.

### 4.3. Surrogate Molecular Subtypes as Determined by Immunohistochemistry

Tumor subtype is defined by hormone receptor and HER2 status, as well as by Ki-67 immunoreactivity, and has extensively been described as a feature that could influence response to NAC [7,9,10,12,61,62].

The CTNeoBC study pooled data from 12 international trials that included 11,955 early BC patients treated with NAC. The more aggressive subtypes were associated with pCR and better long-term outcomes. Those aggressive subtypes were TNBC, HER2-enriched and high-grade HR-positive tumors. These results are similar to the ones obtained by the pooled analysis of the German neo-adjuvant chemotherapy trials conducted by von Minckwitz et al. [62].

More recently, in the study from Haque et al. including 14,000 women, the highest rate of pCR was seen in HER2-enriched subtypes, followed by TNBC and luminal B [10]. In this study, the luminal A subtype had the lowest pCR rate.

The association of the HER2-enriched subtype with pCR was evaluated in the presence of HER2-targeted agents in the NAC regimen but also with mono or dual blockade and different HER2-targeted agents such as pertuzumab, trastuzumab, T-DM1 and lapatinib. In the meta-analysis of Shen et al., pCR was achieved in the HER2-subtype, irrespective of the NAC regimen and targeted therapy used, as long as an HER2 targeting agent was used [61].

Despite the fact that pCR is predictable in the TNBC subtype, TNBC with pCR still has more of a risk of relapse than other subtypes with pCR, which could be explained by the high degree of heterogeneity within the TNBC subtype, in terms of both genomic and transcriptomic profiles.

### 4.4. Tumor-Infiltrating Lymphocytes (TILs)

TILs are evaluated on hematoxylin and eosin slides and can be assessed in the stroma and in the intratumoral area. Stromal TILs are present in the tumor microenvironment without contact with the tumor cells, whereas intratumoral TILs are defined as TILs found in the tumor zone or in the peritumoral area in contact with tumor cells. In breast cancer, stromal TILs evaluation is considered the most reproducible parameter since stromal TILS are more abundant than intratumoral TILs [63,64,65].

The correlation between the levels of TILs and pCR in the neoadjuvant setting has been evaluated in several studies and in all immunohistological subtypes (Table 2). Luminal breast cancer presents fewer TILs than the HER2-enriched and TNBC subtypes.

In early luminal breast cancer, the GeparDuo and GeparTrio randomized clinical trials (RCT) both evaluated TIL levels in patients receiving neoadjuvant chemotherapy with anthracyclines and taxanes. In those studies, higher TIL levels in pre-treatment biopsies were associated with a higher rate of pCR [66]. The same conclusion was drawn in the GeparQuinto trial [67]. In the pooled analysis of the six studies performed by the German Breast Group (*n* = 3771 patients) (GBG), a high percentage of TILs at diagnosis correlated with a higher chance of achieving pCR, but not in the luminal subgroup alone (*n* = 1366) [68].

The HER2-enriched subtype is known to be associated with more TILs in both the stromal and tumor areas. In the HER2-positive cohort of the GeparTrio trial, in which no trastuzumab was given in NAC, a complete response was correlated with higher TILs [66]. In the six studies from the GBG, wherein 1379 HER2-positive tumors were assessed for TILs, the pCR rates differed significantly between high-TILs tumors (> or =60% TILs) and low-TILs tumors (<10% TILs) (49 vs. 32% respectively, *p* < 0.005) [68]. Solinas et al. conducted a meta-analysis of five RCTs evaluating TILs in the neoadjuvant setting of HER2-positive disease, in which trastuzumab was given alone or in combination with lapatinib [63]. The results of this meta-analysis were also in favor of an increased likelihood of achieving a pCR in the presence of higher baseline TIL levels. Moreover, the benefit obtained was independent of the backbone chemotherapy or anti-HER2 agents used. Currently, dual blockade with the combination of trastuzumab and pertuzumab is increasingly administered in high-risk patients. In this context, three clinical trials have evaluated TILs in patients receiving the combination [69,70,71]. Two of the three studies, NeoSphere and GeparSepto, did not show any association between the levels of TILs and pCR. A pooled analysis of these exploratory small subgroups could provide more answers.

In TNBC, lymphocyte infiltrates have been found to be increased compared to other subtypes, with stromal TILs ranging from 15 to 90% and intra-tumoral TILs ranging from 5 to 10% [63,72]. The GeparSixto trial evaluated 314 patient samples and showed that highly infiltrated tumors were associated with pCR [73]. In the GeparQuinto trial, the exploratory analysis of the 104 TNBC patients did not show any significant correlation between TIL and pCR levels [67]. Nevertheless, the correlation between TILs and pCR was confirmed in the meta-analysis of the six studies from the GBG (906 TNBC patients) [68]. Loibl et al. have reported that stromal TILs are associated with higher pCR rates in both cohorts of the GeparNuevo trial, but in contrast, intratumoral TILs were not [74]. Despite all the conducted RCTs and retrospective studies, the data regarding TILs do not yet appear mature enough to be used in clinical practice as a predictive biomarker, although the results are encouraging for both HER2-enriched and TNBC patients [64]. The variability between centers and pathologists remains an important issue to be solved for future trials. Nevertheless, Van Bockstal et al. showed that, despite the inter-observer variability, stromal TILs were significantly correlated with pCR [72,75].

### 4.5. PD-L1 Expression

Breast cancer is considered less immunogenic than other cancer types. Nevertheless, TNBC has been highlighted as the subtype with the highest expression of PD-L1 due to the genomic instability found in this particular subtype [76]. Several studies have evaluated PD-L1 expression in breast cancer, especially in TNBC, with conflicting results concerning the correlation between PD-L1 expression and its predictive value in the neoadjuvant setting [77,78]. These reported conflicting results could be explained by several factors: the heterogeneity of breast cancer itself, the biopsy type (surgical piece vs. needle), the use of different FDA-approved PD-L1 antibodies, and the different methodologies used across studies to evaluate PD-L1 (the consideration of the tumor cells and/or immune cells, the calculation of the combined positive score (CPS) or tumor proportion score (TPS), the use of different cut-offs) [77,79]. In early-stage TNBC, the phase 1 KEYNOTE-173 trial evaluated the addition of pembrolizumab to neoadjuvant chemotherapy. Samples from pre-treatment tumors were stained for PD-L1 with the 22C3 antibody, and the analysis was based on CPS calculation. PD-L1 positivity was associated with increasing rates of pCR when combining the checkpoint inhibitor with common chemotherapy [80]. In the second interim analysis of the phase 3 KEYNOTE-522 study, the pCR rate was significantly higher in the pembrolizumab group, independent of PD-L1 expression. In addition, PD-L1-positive patients had a higher rate of pCR in both arms [81]. The GeparNuevo and Impassion031 studies also reported a higher pCR rate in the PD-L1-positive subgroup in comparison to the PD-L1-negative subgroup [74,82]. In contrast, in the NeoTRIPaPDL1 trial, no difference in pCR rates was observed between the PD-L1-positive and the PD-L1-negative groups [83]. Again, these contradictory results might partly be explained by the relatively high degree of inter-observer variability among pathologists in PD-L1 immunohistochemical assessment, as reported for both primary and metastatic TNBC [84,85].

**Table 2 cancers-14-03876-t002:** Secondary analysis studies evaluating TILs as predictive biomarkers in the neoadjuvant setting.

Trials	Year of TILs Subanalysis	Number of Patients	Number of Patients for TILs Subanalysis	Subtypes (n)	NAC Regimens	pCR Rates
GeparDuo [66,86]	2010	913	218	All	4× doxorubicin + docetaxel q2w (ADoc) vs. 4× doxorubicin + cyclophosphamide and 4× docetaxel q3w (ACDoc)	7% (ADoc) vs. 14% (ACDoc)
GeparTrio [66,87]	2010	2090	840	All	docetaxel + doxorubicin + cyclophosphamide (TAC) vs. vinorelbine + capecitabine (NX)	5.3% (TAC) vs. 6% (NX)
GeparQuattro [67,88]	2016	1509	178	HER2-negative (*n* = 1058) HER2-positive (*n* = 451)	4× epirubicin + cyclophosphamide + 4× docetaxel + trastuzumab +/− capecitabine in HER2 positive 4× epirubicin + cyclophosphamide + 4× docetaxel +/− capecitabine in HER2 negative	31.7% (HER2-positive) vs. 15.7% (HER2-negative)
GeparQuinto [67,89]	2016	615	320	HER2-positive	4× epirubicin + cyclophosphamide + 4× docetaxel + trastuzumab (T) vs. lapatinib (L)	30.3% (T) vs. 22.7% (L)
GeparSixto [73,90]	2015	588	580	HER2-positive (*n* = 273) TNBC (*n* = 315)	In HER2-positive: paclitaxel + doxorubicin + trastuzumab + lapatinib +/− carboplatin In TNBC: paclitaxel + doxorubicin +/− carboplatin +/− bevacizumab	
NeoALTTO [91]	2015	455	387	HER2-positive	Lapatinib (L) vs. trastuzumab (T) vs. lapatinib + trastuzumab (LT)	20% (L) vs. 27% (T) vs. 44% (LT)
CherLOB [92]	2016	121	121	HER2-positive	Paclitaxel + FEC + trastuzumab (T) vs. lapatinib (L) vs. lapatinib + trastuzumab (LT)	25% (T) vs. 26.3% (L) vs. 46.7% (LT)
GeparSepto [71,93]	2017	1206	1206	HER2-negative (*n* = 810) HER2-positive (*n* = 396)	Nab-paclitaxel (nP) or paclitaxel (P) + EC +/− trastuzumab and pertuzumab	38% (nP) vs. 29% (P)
TRYPHAENA [70]	2016	225	213	HER2-positive	Arm A: FEC + trastuzumab + pertuzumab followed by docetaxel + trastuzumab + pertuzumab Arm B: FEC followed by docetaxel + trastuzumab + pertuzumab Arm C: docetaxel + carboplatin + trastuzumab + pertuzumab	61.6% (arm A) vs. 57.3% (arm B) vs. 66.2% (arm C)
NeoSphere [69]	2015	417	350	HER2-positive	Group A: trastuzumab + docetaxel Group B: trastuzumab + pertuzumab + docetaxel Group C: pertuzumab + trastuzumab Group D: pertuzumab + docetaxel	29% (group A) vs. 45.8% (group B) vs.16.8% (group C) vs. 24% (group D)
GeparNuevo [74]	2019	174	171	TNBC	Nab-paclitaxel +/− durvalumab followed by EC	53.4% (durvalumab) vs. 44.2% (placebo)

TILs: tumor-infiltrating lymphocytes; NAC: neoadjuvant chemotherapy; pCR: pathological complete response; ADoc: doxorubicin and docetaxel; ACDoc/TAC: doxorubicin, docetaxel and cyclophosphamide; NX: vinorelbin and capecitabine; T: trastuzumab; L: lapatinib; LT: lapatinib and trastuzumab; P: paclitaxel; nP: nab-paclitaxel.

## 5. Predictive Biomarkers under Investigation

### 5.1. Imaging and Radiomics Biomarkers

Imaging plays an important role in the management of early breast cancer, in the initial staging of the tumor and lymph nodes and in the evaluation of response to NAC. In clinical practice, assessments of response are mostly conducted by ultrasound (US) or magnetic resonance imaging (MRI), the second one being the most sensitive technique to determine the presence of a residual tumor. The prediction of response to NAC by imaging before the completion of the treatment has been proposed but is still subject to debate, as will be discussed below. For this purpose, radiomics bring new horizons with the analysis of data extracted from radiological imaging such as textural variables. Using dedicated algorithms, these could help predict response to treatment [94,95,96,97,98,99]. As a non-invasive tool for predicting response to NAC in breast cancer, radiomics data from magnetic resonance imaging, quantitative ultrasound (QUS) or even ^18^F-fluorodeoxyglucose positron emission tomography/computed tomography (^18^F-FDG PET/CT) show promising results in the evaluation of the outcomes of patients [100,101].

#### 5.1.1. MRI

Functional imaging with diffusion weighted imaging MRI (DWI-RMI) and dynamic contrast-enhanced MRI (DCE-MRI), used in daily practice, can be of help by detecting early changes in the properties of the tumors (e.g., angiogenesis, cellularity) with the extraction of quantitative parameters. These changes in cellularity or angiogenesis have been studied after several cycles of NAC and could be correlated with pCR in small cohort studies [102].

Several studies used artificial intelligence (AI) to analyze the value of multiparametric data from pre-treatment MRI to predict response to NAC. Studies have evaluated gadolinium-T1-weighted MRI images from pretreatment acquisition and have shown that these images could predict the subgroup of patients achieving pCR [103,104,105]. Unfortunately, using gadolinium is not always possible, as it is contra-indicated in cases of renal insufficiency or allergy. Pre-treatment T2-weighted MRI (without contrast) features from tumor cores and margins have also been evaluated in 102 patients by Kolios et al., revealing that non-responding tumors have disorganized structures in comparison to chemo-sensitive tumors. Using machine learning classifiers, the evaluation of the tumor texture in these T2-weighted images had an accuracy of 90% to predict response to NAC [100]. Multiple multivariate machine learning-based models using pre-treatment MRI data have been studied in recent years, such as the model developed by Cain et al., which was able to predict pCR in TNBC and HER2-positive patients with an acceptable discrimination (AUC at 0.707, 95%CI: 0.582–0.833, *p* < 0.002) [106]. Chamming and colleagues analyzed texture features on MRI data before NAC and found that some of them were associated with pCR in TNBC [107]. Another study suggested that, with the parameters from intratumoral and peri-tumoral texture, molecular subtypes could be identified by radiomics [108]. Liu et al. developed a radiomics signature with a combination of images from T2-weighted imaging, diffusion-weighted imaging and contrast-enhanced T1-weighted imaging. The signature itself had an accuracy of predicting pCR of 0.79, while the addition of clinical information (e.g., age, molecular classification, Ki-67 status, stage) to this signature improved the accuracy to an AUC of 0.86. They furthermore validated their models on an external dataset [95].

#### 5.1.2. Quantitative Ultrasound

Compared to MRI, ultrasound imaging has several advantages such as its lower cost, the absence of the injection of exogenous contrast agents and the fact that it is transportable. It is therefore more accessible for the screening and evaluation of all patients. QUS is a technique that extracts characteristics of the physical properties of tissues (e.g., elastography) both in intratumoral and marginal regions. Different studies have evaluated the evolution in the structure of the tumor tissue after treatment by QUS. This technique can detect tumor cell death in response to chemotherapy and, in addition, could predict response to NAC after one-to-four weeks of chemotherapy [109,110,111,112,113,114]. Recently, Taleghamar et al. acquired multiple parametric images with QUS from 181 locally advanced breast cancer patients before NAC [115]. After creating a QUS signature composed of four specific features from intratumoral regions, they could predict response to NAC with a sensitivity of 87% and a specificity of 85%. Osapoetra et al. and Dasgupta et al. have also developed their models based on a texture-derived method with the inclusion of 78 patients and 100 patients, respectively, with encouraging results (88% and 87% of sensitivity, 78% and 81% of specificity) [116,117].

#### 5.1.3. ^18^F-FDG PET/CT

^18^F-FDG PET/CT is a molecular imaging technique used in clinical practice in oncology [118]. In breast cancer, PET/CT is essentially used to screen for distant metastases, but numerous studies from the past decade have described a potential role of PET/CT as an instrument for predicting response to NAC [4,101,119,120,121]. Higher glycolytic activities at diagnosis and significant reductions in the standardized uptake value (SUVmax) of the tracer during NAC have been described as predictive factors of response to NAC, but they are still controversial [118].

Recently, the use of radiomics in PET/CT imaging was able to document intratumor heterogeneity and has provided encouraging new results, but this has, until now, only been described in small cohorts of patients [122,123]. In one study, Li et al. retrospectively studied 100 breast cancer patients and showed that PET/CT features could predict pCR. Moreover, the prediction of pCR by their model showed more pCR in TNBC and HER2-positive patients. Luo et al. combined radiomics features from PET/CT with clinic-pathological information such as the patient’s age and Ki-67 levels on diagnostic biopsies, and they also obtained encouraging results concerning the prediction of pCR after the first two cycles of NAC [124].

In conclusion, these models show the potential of radiomics and machine learning to predict response to NAC in breast cancer on MRI, QUS and even ^18^F-FDG PET/CT. This could be of help to individualize the treatment of patients in clinical practice but needs further validation in prospective trials before any conclusion regarding clinical utility can be drawn.

### 5.2. Plasmatic Biomarkers

#### 5.2.1. Peripheral Blood Cells and Ratios

Systemic inflammation at the time of cancer diagnosis is of interest, as it may reflect tumor-associated inflammation. Moreover, neutrophil and lymphocyte counts have been described as predictors of survival and response to therapy in multiple cancer types [125,126]. The neutrophil-to-lymphocyte ratio (NLR), which is the ratio between the absolute numbers of neutrophils and lymphocytes, has been evaluated in several studies in breast cancer, but its use in clinical practice has not yet been implemented because of contradictory findings [125,126,127,128,129,130]. In 2021, Zhu et al. performed a retrospective study of NLR in 346 patients with BC and concluded that NLR could be an independent predictor of pCR after NAC [131]. A higher NLR was indeed associated with lower pCR. In this study, patients were rigorously selected, and patients with a recent surgery or biopsy or with an autoimmune disease or recent infection were excluded. All selected patients received the same NAC regimen, which was not always the case in previous studies. The threshold value in this study was 1.695 and was determined by ROC curve analyses, which is consistent with previous studies using cut-off values ranging from 1.7–4 [126].

Other ratios such as the pan-immune-inflammation-value score (PIV) have been evaluated in cancer patients. PIV is an immune score that is calculated as follows: neutrophil absolute count × platelet absolute count × monocyte absolute count/lymphocyte absolute count [132]. It has been studied in different cancer types such as melanoma and metastatic colorectal cancer [132,133]. In a study including melanoma patients (*n* = 228), high PIV was independently associated with poorer OS, poorer PFS and resistance to immunotherapy and targeted therapy [133]. The same conclusions were drawn in two studies conducted with 438 and 163 metastatic colorectal patients, respectively [132,134]. In breast cancer, a small cohort study of 57 HER2+ patients, all treated with pertuzumab and trastuzumab in combination with chemotherapy, also demonstrated that higher PIV was associated with worse PFS [135]. Nevertheless, none of these previous studies evaluated PIV as a predictive marker of pCR. In 2021, Sahin et al. conducted a retrospective study with 743 recruited BC patients in order to analyze different peripheral blood cell populations and ratios including NLR and PIV, along with the platelet-to-lymphocyte ratio (PLR) and monocyte-to-lymphocyte ratio (MLR) [136]. Their results indicated that low PIV was associated with better pCR rates and therefore better DFS and OS, with PIV being superior to NLR, PLR and MLR [136].

In conclusion, ratios of peripheral blood cell counts could be interesting, easily accessible and inexpensive predictive markers of response to NAC, but before using them in clinical practice, more validation studies with a rigorous selection of patients need to be conducted, such as prospective multi-centric studies including a higher number of patients.

#### 5.2.2. Liquid Biopsies

Liquid biopsies offer a minimally invasive technique for diagnosis, disease monitoring and the evaluation of the response to treatment. Several components of the tumor can be analyzed with liquid biopsy samples, such as circulating tumor DNA (ctDNA), circulating tumor cells (CTCs) and tumor-educated platelets (TEPs) and exosomes. While TEPs and exosomes are currently studied primarily as diagnostic tools, ctDNA and CTCs show promising results in assessing response to treatment and predicting resistance in early breast cancer [137,138,139,140,141,142].

ctDNA

Several studies have already been performed with the objective of evaluating the role of ctDNA to predict response to NAC in breast cancer [137,143,144,145,146,147]. Nevertheless, the methodology, inclusion criteria and conclusions remain variable, and these studies should therefore be interpreted with caution (Table 3).

Garcia-Murillas et al. were pioneers of the use ctDNA analysis in early breast cancer patients to identify patients with a high risk of relapse [143]. They recruited 55 patients, identified somatic mutations on tumor biopsies and designed ddPCR assays for each detected mutation. They tracked ctDNA presence in serial samples during follow-up. At baseline, ctDNA was found in 69% of the plasma samples and was correlated with more aggressive features such as a high histological grade and ER negativity. Nevertheless, the baseline detection of ctDNA was not predictive of disease-free survival or early relapse. The results from samples collected 2-to-4 weeks after surgery showed that only 19% of patients had residual ctDNA. At this timepoint, the presence of ctDNA was associated with early relapse. Tracking the presence of ctDNA in serial plasma samples after surgery was also predictive of early relapse in patients positive for ctDNA.

In 2017, Riva et al. evaluated 46 early TNBC patients, also with personalized ddPCR probes based on mutations found in tumor biopsies [144]. In this study, plasma samples were collected at four timepoints: before NAC, after one cycle of NAC, before surgery and after surgery. Before NAC, 75% of patients had ctDNA detected, and all patients experienced a decrease in ctDNA during the course of the treatment, except for one patient who had tumor progression during NAC. The results of this study were not in favor of an association between ctDNA detection (at any timepoint) and pCR after NAC. However, patients with a slow decrease in the ctDNA level had shorter disease-free and overall survival.

In the NeoALTTO substudy, which only enrolled HER2-positive patients, ctDNA was also analyzed, but only patients with *PIK3CA* and/or *TP53* mutations were selected for further personalized analysis (69/455) [145]. Samples were analyzed at three timepoints: before NAC, during NAC (at week 2) and before surgery. With this method, ctDNA was found in 41% of patients at baseline, 20% of patients at week 2 and 5% of patients before surgery. As seen by Garcia-Murillas et al., ctDNA detection at baseline was also associated with ER-negative status. In this study, the detection of ctDNA before NAC was associated with a decreasing rate of pCR.

In a small study recruiting 10 patients, in which plasma samples were collected prior to NAC and before each infusion of NAC until 1-to-3 years after the completion of NAC, ctDNA “rapid” clearance was correlated with the probability of reaching a pCR. Once again, patients with residual ctDNA after the initiation of the treatment did not achieve pCR, and in those patients, two patients out of the three had a rapid recurrence (<2 years following the start of NAC) [146]. McDonald et al. came to the same conclusion in a cohort of 22 patients, in which they used a more sensitive technique to detect ctDNA [147].

More recently, in the neoadjuvant I-SPY 2 TRIAL evaluating standard NAC alone or combined with MK-2206 (an AKT inhibitor), the plasma samples of 84 patients were collected at different timepoints: pretreatment (T0), 3 weeks after the initiation of paclitaxel (T1), between paclitaxel and anthracycline treatment (T2) and prior to surgery (T3). Personalized tests were performed on primary tumor sections to identify 16 patient-specific mutations, and these mutations were subsequently searched in the plasma samples. At the pretreatment timepoint (T0), 73% of the patients were ctDNA-positive. The patients who achieved pCR were ctDNA-negative after NAC, and moreover, in the case of no ctDNA clearance at T1, the patients had a higher risk to have residual disease after NAC. After a median follow-up of 4.8 years for survival analysis, this study showed that non-pCR patients positive for ctDNA at T3 have an increased risk of metastatic relapse in comparison with those who were also non-pCR but were ctDNA-negative at T3, who had excellent outcomes [137]. Zhou et al. also evaluated ctDNA in 145 patients and demonstrated that the persistence of ctDNA during the treatment was correlated with non-response to NAC (Residual Cancer Burden of II/III) [148].

Taken together, ctDNA is clearly an interesting tool in the arsenal of predictive biomarkers of response to NAC. Although none of the currently available studies have been able to predict response with baseline ctDNA detection, it appears that the early assessment of ctDNA clearance could be a stratification tool to design escalation or de-escalation studies in early breast cancer.

CTCs

Until now, CTC detection has been performed mainly in the metastatic setting, but recent studies have focused on CTC detection in early breast cancer and its kinetics during NAC and adjuvant therapy [149]. CTCs can be detected in single cells (SC) or in clusters (CC) depending on the technique used. CC-CTCs are mostly evaluated in the metastatic setting. Intuitively, in early breast cancers, clusters may be of the most interest, as they are most likely to progress to metastasis in comparison to single cells, which have to overcome several obstacles to survive and disseminate [150]. However, Reduzzi et al. have demonstrated the limitations of the detection of agglomerates of CTCs in early breast cancer [149]. Although the presence of CC-CTCs was found in small study cohorts, no correlation was found with the likelihood of obtaining a pCR [149,151,152]. Moreover, these studies have shown a decrease in CC-CTCs after surgery and not during NAC [149,152]. The CellSearch system used for the detection of SC-CTCs could detect 20–30% of SC-CTCs in the early breast cancer setting before treatment [151,153,154]. In those studies, SC-CTC positivity was correlated with worse outcomes in terms of OS and DFS [151,154]. Concerning the SC-CTC detection during and after the treatment, the persistence of SC-CTCs after the treatment was correlated with a worse outcome [153]. In conclusion, the predictive value of CTCs is still uncertain and needs to be further explored before being used in clinical practice.

### 5.3. Gene Signatures

Several multigene signature assays have been developed in the last decade. They aim to better classify patients based on their intrinsic subtypes and enhance prognosis evaluation [155].

Some of them have already been validated in clinical practice, essentially in HR-positive and HER2-negative tumors: EndoPredict, Oncotype DX, MammaPrint and PAM50. Their utility in daily routines consists in providing an individual risk assessment of disease recurrence and prognostic information in order to better guide adjuvant therapy selection in early disease. The potential value of these well-known multigene profiles as predictive biomarkers of response to NAC has also been evaluated, with interesting results. Nevertheless, their indication in this setting has not yet been validated in clinical practice (Table 4).

#### 5.3.1. EndoPredict—Molecular Score (MS)

EndoPredict is a 12-gene signature measuring the expression of 8 cancer-related genes, 3 reference genes and 1 control gene. The prognostic value of this signature has been validated, stratifying patients treated with adjuvant endocrine treatment (tamoxifen) into a low or a high risk of recurrence at 10 years [156]. Moreover, the addition of clinical features such as nodal status and tumor size to the EndoPredict score is also a good indicator of late recurrence (EPclin) and can help clinicians to decide if additional treatments are needed in case of high-risk scores. In a comparative, non-randomized analysis of two prospective studies of HR-positive and HER2-negative early breast cancer, this multigene score could predict the chemotherapy benefit [157]. Regarding the NAC setting, only a few studies have shown the feasibility of using the MS score in this indication [158,159,160,161]. Bertucci et al. were the first to retrospectively analyze the EndoPredict signature in a cohort of breast cancer patients (*n* = 553) treated with anthracyclines and taxanes-based NAC. They showed that patients with a high MS score have higher pCR rates than those with a low-risk score [160]. In 2020, Soliman et al. analyzed the expression data from six public datasets from the Gene Expression Omnibus (GEO) database, with a total of 764 ER-positive and HER2-negative patients. These 764 samples were selected based on the availability of clinical information such as pCR, IHC status, FISH status and the NAC scheme administered. They compared the EndoPredict score and the Oncotype DX score (see below) and showed that EndoPredict was a good predictive marker of response to NAC [159]. Mazo et al. also published similar results comparing a new six-gene signature (OncoMasTR) to different prognostic signatures, including EndoPredict, in seven GEO datasets, including four common datasets with Soliman et al. [161] Finally, Dubsky et al. evaluated the EndoPredict score in samples from the ABCSG-34 trial, where patients received either NAC or neoadjuvant endocrine therapy based on menopausal status, HR expression, grade and Ki-67. In this analysis, tumors with a high MS score were more likely to be resistant to neoadjuvant endocrine therapy, whereas low-MS-score tumors did not benefit from NAC [158].

#### 5.3.2. Oncotype DX—Recurrence Score (RS)

The Oncotype DX recurrence score is the result of the relative expression quantification of 21 genes (16 cancer-related genes and 5 reference genes). This score allows for the classification of patients into three categories: low risk, intermediate risk and high risk. The prognostic value of RS was validated in the prospective TAILORx and RxPONDER studies, demonstrating that patients with intermediate risk could be spared adjuvant chemotherapy in addition to endocrine therapy [162,163]. Later, the potential predictive value of the RS was evaluated in several retrospective and prospective studies. Various studies reported that a higher RS was associated with a higher rate of achieving pCR by NAC [164,165,166]. More recently, Morales Murillo et al. also described 26% of pCR in high RS (>30) in comparison to patients with RS from 0–30, where no pCR was achieved [167]. In studies evaluating neoadjuvant hormonal therapy, low RS was associated with a better response in comparison to intermediate- and high-risk RS [168,169,170]. However, in a cohort of non-PCR patients (*n* = 60), Soran et al. found that the RS measured on initial samples was not correlated with the percentage of tumor size reduction [171]. Most of the studies used the threshold of RS below 25 or 30 to conclude that there was no benefit of NAC despite the clinical criteria indication for NAC [165,172,173].

#### 5.3.3. Mammaprint

The Mammaprint assay is a 70-gene signature used in post-menopausal early breast cancer patients. This signature classifies tumors in two groups that are associated with good or poor prognosis based on the recurrence risk at 5 and 10 years. In the prospective MINDACT study, patients with ER-positive and HER2-negative early breast cancer and a low Mammaprint score who received endocrine therapy could safely be spared adjuvant chemotherapy [174]. The use of Mammaprint as a predictive marker of response to NAC has only been evaluated in small exploratory studies. Straver et al. studied 167 patients who received NAC [175]. Of those, 144 (86%) had a high Mammaprint score. Interestingly, no pCR was observed in the low-risk group, while 20% of the high-risk patients achieved a pCR. Later, Glück et al. retrospectively evaluated 437 patients from neoadjuvant trials and came to the same conclusion [176]. Nevertheless, these results need to be carefully interpreted since the initial Mammaprint score was established from tumor specimens from the surgical piece and not from diagnosis biopsies.

#### 5.3.4. PAM50—Prosigna Assay

PAM50 is a 50-gene signature used and validated to identify intrinsic molecular subtypes of breast cancer (luminal A, luminal B, HER2-enriched, basal-like) but also to estimate a Risk of Recurrence (ROR) score capable of classifying tumors into low, intermediate or high risk of distant recurrence [177]. This gene signature was developed in order to evaluate the risk of relapse in patients with HR+ and HER2-negative breast cancer and to evaluate the indication of adjuvant chemotherapy in high-risk cases. In the neoadjuvant setting, Prat et al. studied this assay in core needle biopsy samples to evaluate if it was suitable for core biopsies [178]. They found that the Prosigna assay performed on core needle biopsies was reliable in terms of ROR score and intrinsic subtypes classification. Moreover, the Prosigna could predict response to NAC, which was confirmed by Rodriguez et al. [179]. Later, the phase II PROMIX trial evaluated 150 patients with NAC in combination with bevacizumab [180]. By comparing the baseline and on-treatment biopsies, they discovered that an early change in the intrinsic subtype during NAC may be predictive of pCR and EFS. Ohara et al. showed that the luminal A intrinsic molecular subtype defined by PAM50 was a significant predictor of non-pCR in patients receiving NAC and, conversely, that the immunohistochemistry-defined luminal A subtype was not correlated with a low pCR rate [181]. The intrinsic molecular subtype was furthermore associated with pCR independently of standard clinical variables in a study evaluating 957 patients, including 350 triple negative breast cancer patients [182]. Depending on the intrinsic subtypes, different pCR rates were achieved, ranging from 9.3% to 14.2%, 20% and 50% (luminal A, luminal B, HER2-enriched and basal-like, respectively) [179]. More recently, the Prosigna assay was used in a trial including 43 IHC-defined luminal B patients receiving a combination of anthracyclines and anti-PD1 antibodies. The tumors with the basal-like intrinsic subtype had better response to this combination [183].

## 6. Future: Patients-Derived Tumor Organoids (PDTO)

Organoids are defined as 3D in vitro models generated from in vivo tissue or organ that functionally and architecturally recapitulate the original tissue. The 3D culture is established from digested tissue and cultured in suspension either in a hydrogel matrix that mimics the extracellular matrix (ECM) or in an “air-liquid interface”. Culturing processes, including passaging and medium composition, can drastically differ between tissues [184,185].

In cancer research, organoids, and, more specifically, patient-derived tumor organoids (PDTO), are a promising tool since they replicate the heterogeneity but also cell interactions of the tumors [186]. Until today, cancer research has been dominated by other culture models such as 2D immortalized cell lines and patient-derived xenografts (PDX) (Table 5).

Cell lines have been used in the development of drugs and are still used in basic cancer research, which could be explained by the ease of maintenance, the lower cost and the ease of genetic manipulations or use in high-throughput drug screening. Nevertheless, 2D cultures present several limitations. While the maintenance of cell lines is less time-consuming than that of PDX models, the establishment of permanent cell lines from tumors is poorly efficient within breast cancer, with a success rate estimated between 1 and 10%. Furthermore, these cell lines do not represent tumor diversity, since in vitro adaptation modifies heterogeneity by selecting the surviving clones [184]. Moreover, considering the fact that metastatic tumors have more of an ability to generate cell lines than primary tumors or normal tissue, the available immortalized cell lines hardly represent the entire spectrum of cancer disease. Even if co-culture is feasible, the culture conditions of cell lines cannot recapitulate the environment and cell–cell interactions of in vivo tumors and lack other cell types usually present in the neighborhood of the tumor such as fibroblasts or immune cells [187]. The lack of heterogeneity may thus misestimate the biological process as compared to in vivo models.

On the other hand, PDX models allow for the study of transplanted tumors in mice with cells that can be engrafted in a physiological environment but from a different species. The success rate of establishment of PDX is higher than that of cell lines but remains notoriously difficult in breast cancer. The disadvantages of this model are that it is time-consuming and costly in comparison to 2D cultures [184,188]. Xenografts can more accurately recapitulate drug response or response to radiation in tumors, even if these responses are mediated by the host environment. PDX models can thus be used to better study drug response but are not amenable for high-throughput screening.

Between 2D cell lines and PDXs, PDTOs are certainly a promising new tool for the study of drug response but also for disease modeling, gene editing and high-throughput drug screening, owing to their capacity for long-term expansion similar to that of cells in 2D cultures. Compared to xenograft models, PDTOs can be easily cultured and are also less costly and time-consuming [184,189,190,191]. Despite the lack of vascularization and interaction with the immune system, drug sensitivity prediction has proven to be correct in numerous studies evaluating PDTO from different origins [192].

In breast cancer, Sachs et al. established 95 breast cancer organoid cell lines out of 155 tumors, with the majority of organoids matched with the original breast tumor regarding histopathology and hormone/HER2 receptor status. Moreover, the gene expression analysis in organoids appears to be less confounded by variation given the absence of normal cells. This, in theory, also allows for the easier detection of somatic mutations. Out of the 95 lines, 28 were tested with 6 drugs targeting the HER signaling pathway. Responses were seen in models overexpressing HER2. Since the PDTO models were derived from surgical pieces, it was not possible to follow, in parallel, the response to treatment in patients. Nevertheless, xenograft models were generated from one HER2-sensitive PDTO-line and one HER2-resistant PDTO-line. The in vivo and in vitro responses to afatinib were similar [189]. More recently, Guillen et al. established paired PDX and PDX-derived organoids (PDxO). In total, 40 PDxOs were derived from PDX out of 47 attempts. The initial validation involved a comparison of matched tumors, PDXs and PDxOs, which revealed that PDxO models were similar to the corresponding primary tumors in 11 models and that, overall, the mutational drivers were stable despite multiple passages of the PDxO lines. Moreover, the drug responses in PDxOs models were concordant with the responses in PDXs models, suggesting that PDxOs could predict response in PDXs correctly. After receiving a biopsy from an early TNBC and the derivation of PDXs and PDxOs, in vivo drug screening performed at clinical relapse demonstrated resistance to 5-fluorouracil, cabozantinib and the NAC regimen (doxorubicin, cyclophosphamide, paclitaxel), as was seen in the clinical setting. The sensitivity to eribulin noted in both models was also present in the clinical setting for a period of nearly 5 months [193]. Given this concordance between clinical and PDX/PDxO responses, high-throughput drug screening in organoids could help to identify resistance and guide treatment decisions in the future. Up to now, the majority of clinical trials involving PDTO have been observational, comparing drug screening in vitro and clinical response (Table 6) [194]. In breast cancer, PDTO models are also of interest as direct functional assays to assess PARP inhibitors’ sensitivity [195]. As discussed by Morice et al., next-generation sequencing (NGS) can fail to identify patients who could benefit from PARP inhibitors in several cancer types (ovarian, breast, pancreatic). In the same way, Sachs et al. have tested four of their breast-PDTO models with the PARP inhibitors olaparib and niraparib. Two models out of the four were sensitive and presented a high *BRCA1/2* signature, despite the fact that one of the sensitive models was not *BRCA1/2*-mutated, showing one limitation of NGS [189]. Pauli et al. have described the creation of a robust precision medicine platform associating whole exome sequencing (WES) and high-throughput drug screening after reporting that only 9.6% of analyzed advanced cancer patients had potentially targetable gene alterations [196]. After collecting 145 tumor samples from several origins (e.g., prostate, colorectal, breast, etc.), 56 PDTO lines were successfully established. PDTO lines have been used for further drug screening with drug libraries combining ~160 drugs (chemotherapeutics and FDA-approved targeted agents). Based on the best results obtained with PDTO, PDX models were also screened with these drugs in monotherapy or in combination. In two cases of gynecological cancer harboring the same mutation in *PIK3CA* and *PTEN*, the drug response profiles were completely distinct, showing that, used together, PDTO and NGS/WES could better select patients who could benefit from these therapies [195].

Assays-guided treatment trials have been introduced in colorectal cancer patients, aiming to investigate the sensitivity of PDTO to different drugs and, depending on the results, to guide the treatment of patients in a personalized manner. The SENSOR trial included 54 patients, from whom 31 organoid lines were derived [197,198]. Of the 25 PDTO-lines tested for 8 selected drugs, 19 responded to at least one drug. A total of six patients received treatment based on PDTO response, but no durable clinical response was observed. The APOLLO trial successfully derived organoids from peritoneal metastases in 19 colorectal cancer patients out of the 28 attempts [199]. Drug screening allowed for the adaption of the systemic treatment in two multi-resistant patients. Despite these first translational encouraging data, this culture model still remains in its infancy, notably regarding breast cancer. The highly variable success rate (ranging from 30 to 90%) and the delays (from 10 days to 2 months) required to establish the models, as well as the non-standardized experimental culturing methods for drug-screening, remain important limitations concerning PDTO cultures that do not allow for the scale-up to large clinical trials [200]. Nevertheless, PDTOs remain an interesting model offering an optimistic perspective for the future in terms of personalized treatment and the prediction of response [201] (Figure 3).

## 7. Conclusions

Predicting the response to NAC in early breast cancer still needs dedicated investigations, since most of the studies performed until now only considered one parameter, limiting their performances. This field remains an area of unmet clinical need, as exemplified by triple negative early breast cancer, where neoadjuvant escalation strategies have recently changed the treatment landscape. In the recently published Keynote-522 trial, the NAC backbone contained carboplatin in both treatment arms, and the addition of pembrolizumab led to a significantly higher pCR rate (64.8 vs. 51.2%) compared to placebo. Nevertheless, in the patients achieving pCR, recurrence rates were not significantly different between the treatment groups [202]. Thus, 50% of the patients do not require the addition of immunotherapy to chemotherapy. As treatment side effects were more pronounced in the more heavily treated patient population, finding a biomarker predictive of response to chemotherapy would be clinically and economically useful. At the same time, a better selection of patients for NAC would avoid directly ruling out promising new agents but also avoid the emergence of resistant clones due to prolonged drug exposure [8]. For a better selection of patients, for developing new drugs and avoiding the residual disease, it is therefore essential to explore and develop new predictive biomarkers with high sensitivity and specificity.

Furthermore, further analyses of these biomarkers could also provide insights into more effective and rational treatment escalation trials. Promising techniques such as radiomics data, liquid biopsies and their combination with more common clinical and histological features may be of great help in deriving effective new biomarkers of response to NAC and circumventing the inter- and intra-heterogeneity that characterize breast cancer.

## Figures and Tables

**Figure 1 cancers-14-03876-f001:**
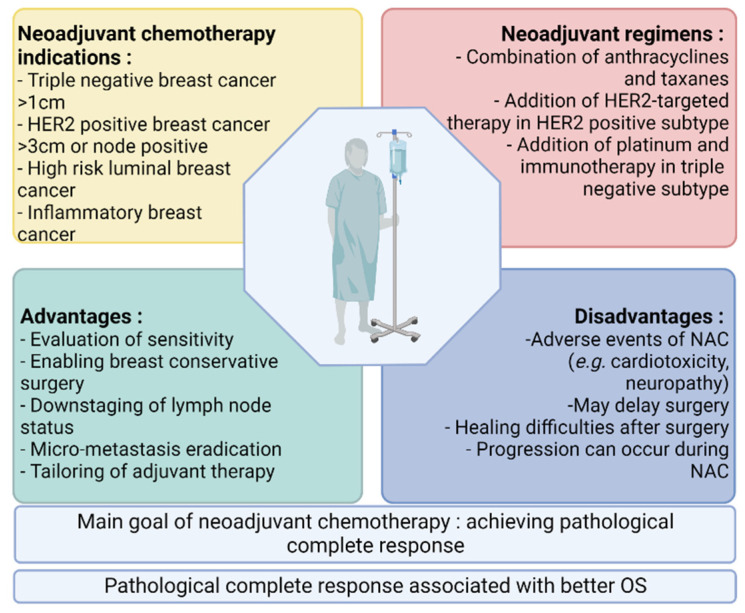
Overview of current indications, regimens, advantages and disadvantages of NAC.

**Figure 2 cancers-14-03876-f002:**
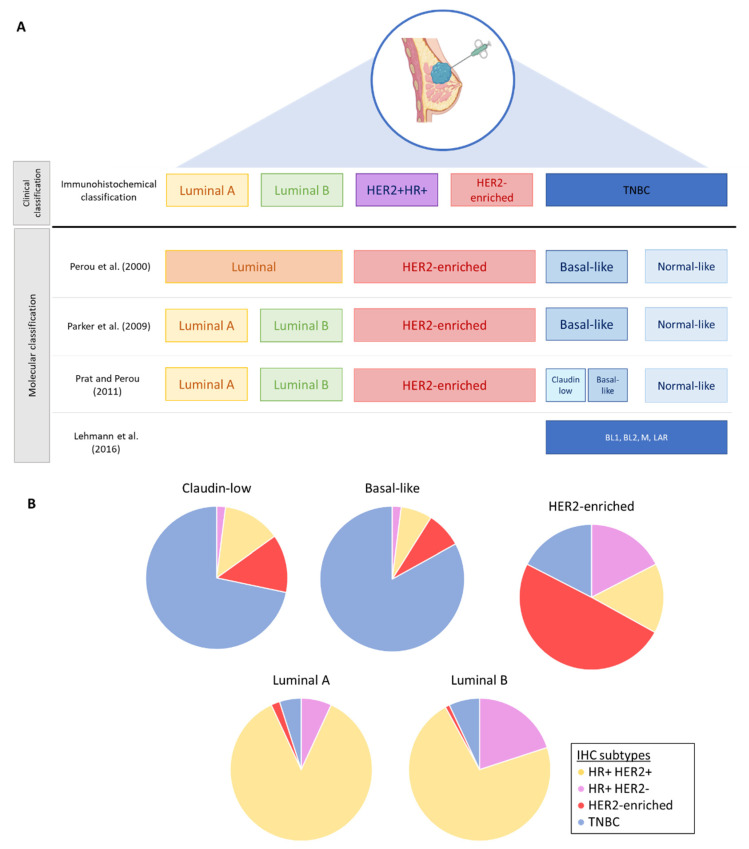
Breast cancer subtypes and intrinsic classification. (**A**) Comparison between immunohistochemical classification and molecular classification [17,18,24,25]. (**B**) Distribution of immunohistochemical subtypes in the molecular classification, as defined by Prat and Perou [24]. HR+: hormone receptor-positive; TNBC: triple negative breast cancer; BL1: basal-like 1; BL2: basal-like 2; M: mesenchymal; LAR: luminal androgen receptor.

**Figure 3 cancers-14-03876-f003:**
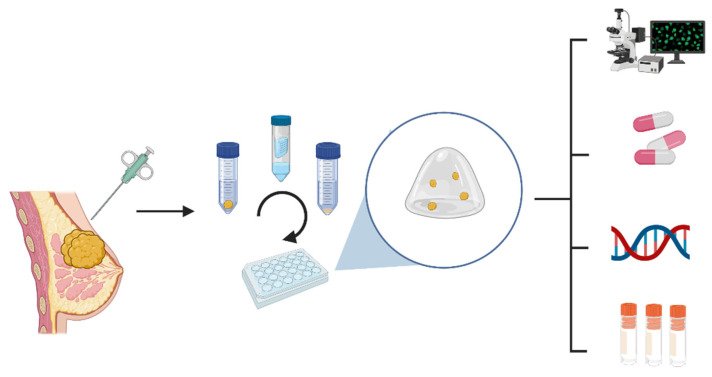
Organoids processing. After collecting fresh tumor tissue, the sample is digested enzymatically and seeded in a gel dome. After several weeks of growing, this model can be used for different purposes, such as: staining characterization, drug-screening, genomic characterization and biobanking.

**Table 1 cancers-14-03876-t001:** Drugs used in NAC regimens in early breast cancer, indications and side effects. AC: doxorubicin and cyclophosphamide; EC: epirubicin and cyclophosphamide; CMF: cyclophosphamide, methotrexate, 5-fluorouracile; TC: docetaxel and cyclophosphamide.

Breast Cancer Subtype	NAC Backbone	Drug Added to NAC Backbone	Indications	Side Effects
**Luminal B**				
	Sequential AC or EC—taxanes		Hormone-receptor-positive cancers larger than 2 cm and/or with axillary lymph node involvement	Cardiotoxicity, hair loss, peripheral neuropathy, febrile neutropenia, fatigue, nausea, diarrhea
	CMF		In elderly patients	Hair loss, peripheral neuropathy, febrile neutropenia, fatigue, nausea, diarrhea, hand-foot syndrome
	TC		If at risk of cardiac complications	Hair loss, peripheral neuropathy, febrile neutropenia, fatigue, nausea, diarrhea
**HER2-positive**				
	Sequential AC or EC—taxanes	Trastuzumab	Node-negative	Chemotherapy side effects: Cardiotoxicity, hair loss, peripheral neuropathy, febrile neutropenia, fatigue, nausea, diarrhea Trastuzumab side effects: Transient cardiotoxicity, diarrhea
	Sequential AC or EC—taxanes	Trastuzumab and pertuzumab	Node-positive	Chemotherapy side effects: Cardiotoxicity, hair loss, peripheral neuropathy, febrile neutropenia, fatigue, nausea, diarrhea Trastuzumab and pertuzumab side effects: Transient cardiotoxicity, diarrhea, peripheral neuropathy
**TNBC**				
	Sequential AC or EC—Carboplatin and taxanes	Pembrolizumab		Chemotherapy side effects: Cardiotoxicity, hair loss, peripheral neuropathy, febrile neutropenia, fatigue, nausea, diarrhea Pembrolizumab side effects: Cutaneous, endocrinopathy, cardiotoxicity, diarrhea, inflammatory pneumopathy, arthritis, hepatitis, nephritis

**Table 3 cancers-14-03876-t003:** Studies evaluating ctDNA in early breast cancer.

Authors	Year	N	Subtypes	Timepoint			
				Before NAC	During NAC	Before Surgery	After Surgery
Garcia-Murillas et al. [142]	2015	55	All subtypes	Yes			Yes
Riva et al. [143]	2017	46	TNBC	Yes	Yes	Yes	Yes
Rothé et al. [144]	2019	69	HER2-positive	Yes	Yes	Yes	
Butler et al. [145]	2019	10	All subtypes	Yes	Yes	Yes	Yes
McDonald et al. [146]	2019	22	All subtypes	Yes	Yes	Yes	
Magbanua et al. [136]	2021	84	All subtypes	Yes	Yes	Yes	
Zhou et al. [147]	2021	145	HR+ and TNBC	Yes	Yes	Yes	

**Table 4 cancers-14-03876-t004:** Available gene signatures and their current indications.

Gene Signature	Number of Genes	Genes	Validated Indications	Utilization
EndoPredict (MS)	12	*BIRC5, UBE2C, DHCR7, RBBP8, IL6ST, AZGP1, MGP, STC2, CALM2, OAZ1, RPL37A*	Evaluation of recurrence at 5–10 years	Score range from 0 to 15 <5: low risk ≥5: high risk
Oncotype DX (RS)	21	*CCNB1, MYBL2, MMP11,* *CTSL2, GRB2, ESR1, PGR, BCL2, BAG1, Ki-67,* *ACTB,* *GAPDH, RPLPO, GUS, TRFC, STK15, BIRC5, HER2, SCUBE2, GSTM1, CD68*	Evaluation of 10-year recurrence in patients	Score range from 0 to 100 (TAILORx) <11: low risk 11–25: intermediate risk >25: high risk
Mammaprint	70	*BBC3, EGLN1, TGFB3, ESM1, IGFBP5, FGF18,* *SCUBE2, TGFB3, WISP1, FLT1, HRASLS, STK32B, RASSF7, DCK, MELK, EXT1, GNAZ, EBF4, MTDH, PITRM1, QSCN6L1, CCNE2, ECT2, CENPA, LIN9, KNTC2, MCM6, NUSAP1, ORC6L, TSPYL5, RUNDC1, PRC1, RFC4, RECQL5, CDCA7, DTL, COL4A2, GPR180, MMP9, GPR126, RTN4RL1, DIAPH3, CDC42BPA, PALM2, ALDH4A1, AYTL2, OXCT1, PECI, GMPS, GSTM3, SLC2A3, FLT1, FGF18, COL4A2, GPR180, EGLN1, MMP9, LOC100288906, C9orf30, ZNF533, C16orf61, SERF1A, C20orf46, LOC730018, LOC100131053, AA555029_RC, LGP2, NMU, UCHL5, JHDM1D, AP2B1, MS4 A7, RAB6B*	Early and distant relapse	Low risk High risk
PAM50—Prosigna	50	*UBE2C, PTTG1, MYBL2, BIRC5, CCNB1, TYMS, MELK, CEP55, KNTC2, UBE2T, RRM2, CDC6, ANLN, ORC6L, KIF2C, EXO1, CDCA1, CENPF, CCNE1, MKI-67, CDC20, MMP11, GRB7, ERBB2, TMEM45B, BAG1, PGR, MAPT, NAT1, GPR160, FOXA1, BLVRA, CXXC5, ESR1, SLC39A6, KRT17, KRT5, SFRP1, BCL2, KRT14, MLPH, MDM2, FGFR4, MYC, MIA, FOXC1, ACTR3B, PHGCH, CDH3, EGFR*	-Risk of Recurrence Score (ROR) -Relapse at 10 years	-Risk of recurrence: low, intermediate, high -Relapse at 10 years in %

**Table 5 cancers-14-03876-t005:** Comparison between cell lines, PDTO models and PDX models.

Features	Cell Lines	PDTO	PDX
Establishment	+	++	++
Maintenance	+++	+	−
Heterogeneity	−	+	++
Patient-specific	−	+++	+++
Environment interactions	−	−	+++
Preservation of tissue feature	−	++	+++
Co-culture	+	+	++
Genetic manipulation	+++	++	−
High-throughput screening	+++	+++	−
Cost	+	++	+++
Time-consuming	+	++	+++
Expertise	+	+++	+++

−: less likely; + to +++: likely to highly likely.

**Table 6 cancers-14-03876-t006:** Clinical trials investigating PDTO models in breast cancer.

Studies	Status	Type of Study	Aim
NCT04450706	Recruiting	Interventional	Treatment decision based on genome sequencing (blood) and drug screening on organoids
NCT05177432	Recruiting	Interventional	QPOP-based drug screen assay to select patients for therapy
NCT04727632	Recruiting	Interventional	Evaluation of the use of [18F] Fluoroestradiol (FES)-PET/CT imaging and the correlation of the results with the drug profiling conducted in organoids
NCT04531696	Recruiting	Interventional	Post-mortem tissue donation program with multi-level and multi-region sample analysis to unravel metastatic breast cancer evolution, biology, heterogeneity and treatment resistance
NCT04281641	Recruiting	Interventional	Evaluation of the correlation between early changes in multiple markers and pathological complete response in patients with HER2-positive breast cancer receiving carboplatin, docetaxel and trastuzumab plus pertuzumab (TCHP) pre-operatively. Markers are examined by gene expression assays, 18F-FDG-PET, 68 Ga-Affibody HER-2 Imaging PET and organoid drug sensitivity
NCT02732860	Recruiting	Observational	Personalized patient-derived xenografts (pPDX) and organoids for drug screening
NCT04703244	Recruiting	Observational	Generate PDX and PDTO models from residual tumors after NAC for drug testing and the study of mechanisms of resistance
NCT03896958	Recruiting	Observational	Establish a data and tissue biobank
NCT05134779	Recruiting	Observational	Live biobank study with samples collected at inflection points in the course of the disease (at the time of initial diagnosis, at the time of surgery and during recurrence or metastasis)
NCT04723316	Recruiting	Observational	Create a national framework with molecular profiling of circulating tumor DNA and/or tumor tissue (optional)
NCT04526587	Recruiting	Observational	Investigate the clinical course of CDK4/6 inhibitor-treated patients in the real-world setting (cfDNA, organoids, PDX models)
NCT05007379	Not yet recruiting	Observational	Test the new CAR-macrophages drug on PDTO
NCT04504747	Not yet recruiting	Observational	Establishment of PDTO models from tumors exposed to NAC in parallel with the study of CTCs, along with tumors before and after NAC, to better identify mechanisms of resistance
NCT05317221	Not yet recruiting	Observational	Study of the heterogeneity and mechanisms of resistance
NCT05381038	Not yet recruiting	Interventional	QPOP drug selection followed by CURATE.AI-guided dose optimization for azacitidine combination therapy (docetaxel or paclitaxel or irinotecan)
NCT04655573	Not yet recruiting	Observational	Assess the feasibility of generating patient-derived micro-organospheres (PDMO) and drug screening

## Data Availability

Not applicable.

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
