# Peer review of "Predictive Biomarkers of Response to Neoadjuvant Chemotherapy in Breast Cancer: Current and Future Perspectives for Precision Medicine"

_cancers, 2022, doi:10.3390/cancers14163876_

Round 1

Reviewer 1 Report

Breast cancer was well identified pathologies and developed many therapeutic approaches on physical, chemical, biological fields. In research area, many colleges were investigated new leading compounds and approved as new anti-breast cancer drugs with long time whereas many cancer-related medicinal doctors initiates clinical lab data-based classical chemotherapies to almost breast cancer patients before wait new techniques.

Neoadjuvant chemotherapy is useful protocol in breast cancer and physician should predict the responses daily with clinical lab data too. In this review, authors were arranged prognostic- and perspective-knowledges to neoadjuvant chemo-related responses in breast cancer patients. Moreover authors suggested also drug screening after individual cell characterization with biobanking systems.

In minor comments, authors needs to describe previously progressed trials such as organoid-based anti-BrCA drug screening from chemical library, hypothetical-, synthetic-compounds and etc. Because these comments were widely propagated and raised availabilities in similar research field

Reviewer 2 Report

Derouane F et al presented a comprehensive review on biomarkers with regard to neoadjuvant chemotherapy in breast cancer. It has a great potential  to be implemented in daily practice. Once it is published, it will attract general interests from our readers in breast cancer field. In addition, it provides up-to-date overview of pathological markers, genetic signatures, radiological techniques, and liquid biopsies. All these important and timely updates will bridge the gap of knowledge between the cutting-edge research and clinical protocols. I enjoyed reading this review.

Reviewer 3 Report

This manuscript focuses on the tumoral phenotypic heterogeneity in breast cancers and the emerging techniques and preclinical studies that restrict neoadjuvant systemic therapy in breast cancers. The manuscript is well-written. Prior to publishing, I have some suggestions to further improve the manuscript; the Quality of Figures is acceptable. See the comments below:

1.      Authors are advised to mention the drawbacks or challenges of neoadjuvant systemic therapy strategies for targeted delivery of therapeutic cues to cope with the resistant types of breast cancers to make the background and discussion stronger.

2.      It is better to include a short paragraph on the adverse effects of neoadjuvant systemic therapy in the manuscript.

3.      It would be nice to compare and contrast about different types of therapeutics and put a table about that.

4.      In the manuscript, the following references may be considered:

DOI: 10.3390/ijms22094597

DOI: 10.1016/j.ijbiomac.2022.03.057

5.      Author should ensure that he/she had the permission of using images from other resources.

Round 2

Reviewer 3 Report

The authors addressed all my concerns.